# MODEL2SCENE: LEARNING 3D SCENE REPRESENTATION VIA CONTRASTIVE LANGUAGE-CAD MODELS PRE-TRAINING

## ABSTRACT

Current successful methods of 3D scene perception rely on the large-scale annotated point cloud, which is tedious and expensive to acquire. In this paper, we propose Model2Scene, a novel paradigm that learns free 3D scene representation from Computer-Aided Design (CAD) models and languages. The main challenges are the domain gaps between the CAD models and the real scene's objects, including model-to-scene (from a single model to the scene) and synthetic-to-real (from synthetic model to real scene's object). To handle the above challenges, Model2Scene first simulates a crowded scene by mixing data-augmented CAD models. Next, we propose a novel feature regularization operation, termed Deep Convex-hull Regularization (DCR), to project point features into a unified convex hull space, reducing the domain gap. Ultimately, we impose contrastive loss on language embedding and the point features of CAD models to pre-train the 3D network. Extensive experiments verify the learned 3D scene representation is beneficial for various downstream tasks, including label-free 3D object salient detection, label-efficient 3D scene perception and zero-shot 3D semantic segmentation. Notably, Model2Scene yields impressive label-free 3D object salient detection with an average mAP of 46.08% and 55.49% on the ScanNet and S3DIS datasets, respectively. The code will be publicly available.

Visual perception on 3D point clouds is fundamental for autonomous driving, robot navigation, digital cities, etc. Although the current fully-supervised methods yield impressive performance, they rely on the large-scale annotated point cloud, which is tedious and expensive to acquire. Moreover, most methods are domain-specific, *i.e.*, a neural network performs well in restricted scenarios with a similar distribution to the training dataset but fails to handle other scenarios with large domain gaps. Therefore, there is an urgent need for an efficacy method to reduce the amount of data annotation and have good cross-dataset generalization capabilities.

Some current methods are approaching the above issues as a domain adaptation problem. Typically, the neural networks are trained on the annotated source datasets and are expected to perform well on the target datasets with significant domain gaps. However, they still require labour-expensive point-level annotation of source domain data for supervision. Other efforts develop self-supervised methods to handle the above issues, *e.g.*, they contrastively learn the positive and negative point pairs to pre-train the network to achieve superior performance with limited annotated data. However, they suffer from an optimization conflict issue, which hinders representation learning, especially for scene understanding. For example, two randomly sampled points in a scene are probably on the same object with the same semantics, *e.g.*, floor, wall and large objects. The contrastive learning process tends to separate them in feature space, which is unreasonable and will harm the downstream task's performance (Chen et al., 2023; Sautier et al., 2022).

To address the abovementioned issues, we propose Model2Scene, a novel paradigm that learns 3D scene representation from Computer-Aided Design (CAD) models and languages. We conclude that two main domain gaps hinder knowledge transferring from CAD models to 3D scenes. One is the model-to-scene gap, *i.e.*, the CAD models are independent and complete, while the objects in a scene have diverse poses, sizes and locations and are obscured by other objects. Another is the synthetic-to-real gap. For example, the surface of CAD models is clean and smooth, while the objects in a real scan are irregular and noisy due to the scanning equipment. Based on the observation,

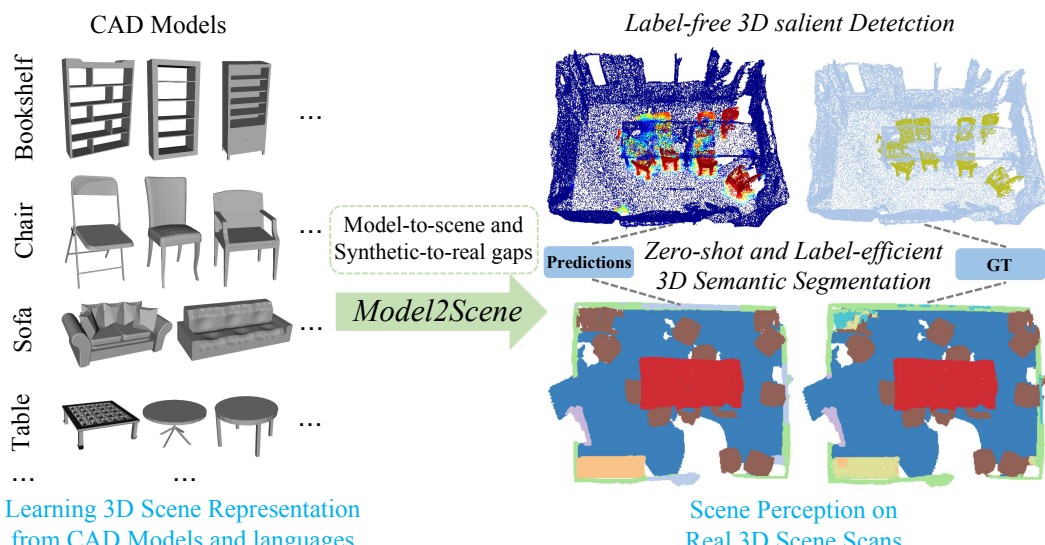

Figure 1: We propose Model2Scene, which learns 3D scene representation from CAD models and languages. Model2Scene emphasize solving the gaps of model-to-scene (from a single model to the scene) and synthetic-to-real (from synthetic model to real scene's object) between the CAD model and the real scene's objects. The learned 3D scene representation is beneficial for label-free 3D salient detection, zero-shot and label-efficient 3D semantic segmentation.

Model2Scene is carefully designed in terms of data pre-processing, latent space regularization and objective function. In data pre-processing, we mix CAD models to simulate a crowded scene that reduces the model-to-scene gap. For latent space regularization, we draw inspiration from the convex hull theory (Rockafellar, 2015), *i.e.*, any convex combinations of the points must be restrained in the convex hull. In light of this, we propose a novel feature regularization operation, termed Deep Convex-hull Regularization (DCR), to project point features into a unified convex hull space that further reduces the domain gap between the CAD models and the real scene's objects. Lastly, we introduce language semantic embeddings as anchors for contrastive learning of points in CAD models. The points on the same CAD models are pulled together in feature space, while the points on the different CAD models are pushed away.

To this end, Model2Scene exhibits the following properties that intuitively address the drawbacks of the previous method. Firstly, compared to dense annotations on the large-scale scene, the label of instance 3D object is easy and convenient to obtain. Secondly, by introducing the entire 3D object, we have the explicit point clusters information to avoid the optimization conflict issue, *i.e.*, those points in the same instance are unreasonably pushed away in feature space. Lastly, the point visual feature is aligned with language semantic embedding. Thus, the network is capable of zero-shot ability and can perceive the unseen object.

We conduct the experiments on ModelNet (Wu et al., 2015), ScanNet (Dai et al., 2017), and S3DIS datasets (Armeni et al., 2017), where ModelNet provides the labelled CAD models for training, and the ScanNet and S3DIS provide real scene scans for evaluation. Model2Scene achieves label-free 3D object saliency detection with the average mAP of 46.08% and 55.49% on the ScanNet and S3DIS datasets. Besides, it can be a potential pretext task to improve the performance of the downstream tasks for 3D scene perception. Furthermore, Model2Scene also present a preliminary zero-shot ability for unseen objects.

The contributions of our work are as follows.

- We propose Model2Scene, a novel paradigm that learns 3D scene representation from CAD models and languages.
- We propose a novel Deep Convex-hull Regularization to handle the domain gaps between the CAD models and the real scene's objects.
- Our method achieves promising results of label-free 3D object salient detection, label-efficient 3D perception and zero-shot 3D semantic segmentation.

# 1 RELATED WORK

**Scene Perception on Point Cloud** Point cloud, as a 3D data representation, has been used extensively in various 3D perception tasks, such as 3D segmentation (Kong et al., 2023a;b; Xu et al., 2023; Ouaknine et al., 2021; Chen et al., 2021; Zhu et al., 2021; Cui et al., 2021; Hong et al., 2021; Liu et al., 2023a; Tang et al., 2020; Schult et al., 2022; Liu Youquan, 2023), 3D detection (Contributors, 2020; Li et al., 2021; Zhang et al., 2020; Qi et al., 2019; Zhu et al., 2020; 2019) and registration (Yuan et al., 2021; Lu et al., 2021; Zeng et al., 2021; Chen et al., 2020a). Although promising performance have been achieved, they are trained on the large-scale annotated point cloud, which is tedious and expensive to acquire. Besides, most of them perform well in restricted scenarios with a similar distribution to the training dataset but fail to handle other scenarios with large domain gaps. In this paper, we propose a novel paradigm that learns 3D scene representation from Computer-Aided Design (CAD) models and languages, reducing the amount of data annotation and having good cross-dataset generalization capabilities.

**Transfer Learning in 3D** Transfer learning has been widely employed in various deep learning-based tasks. The main purpose is to improve neural networks' performance and generalization ability under limited annotated data. In 3D scenarios, transfer learning becomes much more critical due to the difficulty of acquiring 3D labelled data. Generally, deep transfer learning includes but is not limited to the following categories: Self-supervised learning that pre-train the network with extra dataset, and fine-tune on the downstream tasks (Chen et al., 2023; Liu et al., 2023b; Mahajan et al., 2018; Xie et al., 2020; Hou et al., 2021; Rao et al., 2022; Yao et al., 2022; Rozenberszki et al., 2022; Kobayashi et al., 2022; Jain et al., 2021; Nunes et al., 2022; Zhang et al., 2021b; Mahmoud et al., 2023; Chen et al., 2020a); Domain adaptation between the source domain and target domain (Saenko et al., 2010; Wang et al., 2020; Qin et al., 2019; Tzeng et al., 2017; Jaritz et al., 2020; 2022; Saltori et al., 2022; Yi et al., 2021; Peng et al., 2021; Cardace et al., 2023); zero-shot learning that trains on the seen classes and is able to recognize the unseen classes of objects (Chen et al., 2022b; Lu et al., 2023; Michele et al., 2021); and few-shot/semi-supervised learning with few annotated data (Yu et al., 2020; Zhao et al., 2021; Wang et al., 2021; Jiang et al., 2021; Chen et al., 2022a; Cheng et al., 2021; Deng et al., 2022; Hou et al., 2021; Jiang et al., 2021). Compared with the above transfer learning methods, our problem setting refers to 3D synthetic models for supervision and inferring the objects on 3D real scenes. Besides, there are some methods (Yi et al., 2019; Avetisyan et al., 2019; Gupta et al., 2015) leverages synthetic objects for learning in real scenes. However, they require scene annotation for supervision. While our method only learns from the labelled synthetic models. Some optimization-based methods (Knopp et al., 2011; Lai & Fox, 2010; Kim et al.; Ishimtsev et al., 2020; Song & Xiao, 2014; Litany et al., 2017; Li et al., 2015; Nan et al., 2012) that fit 3D synthetic models to the real scene's objects for reconstruction, object replacement and registration are out of the scope of our intention. Unlike the above methods, we study the neural network's model-to-scene and synthetic-to-real generalization ability, which transfer knowledge from 3D CAD models to real scenes for the 3D scene understanding.

**3D Data Augmentation** Data augmentation, as a fundamental way for enlarging the quantity and diversity of training datasets, plays an important role in the 3D deep learning scenario, which is notoriously data hungry. Recently, several attempts have been made on designing new 3D data augmentation schemes and studying 3D data augmentation techniques in systematic ways. PointAugment (Li et al., 2020) proposes a learnable point cloud augmentation module to make the augmented data distribution better fit with the classifier. PointMixup (Chen et al., 2020b) extends Mixup (Zhang et al., 2017) augments the data by interpolating between data samples. PointCutMix (Zhang et al., 2021a) further extend Mixup strategy and perform mixup on part level. Mix3D (Nekrasov et al., 2021) creates new training samples by combining two augmented scenes. Inspired by the above methods, we simulate crowded scene for CAD models to cover the diversity of the objects in real unseen scenes.

**Prototype-based Networks** The Prototype-based Memory network has been applied to various problems. NTM (Graves et al., 2014) introduces an attention-based memory module to improve the generalization ability of the network. Gong et al. (2019) adopt a memory augmented network to detect the anomaly. Prototypical Network (Snell et al., 2017) utilize category-specific memories for few-shot classification. Liu et al. (2019) and He et al. (2020) solve the long-tail issue with the prototypes. In this paper, we adopt the learnable prototypes as the support points to formulate a convex hull that alleviates the domain gap between the CAD model and the real scene's objects.

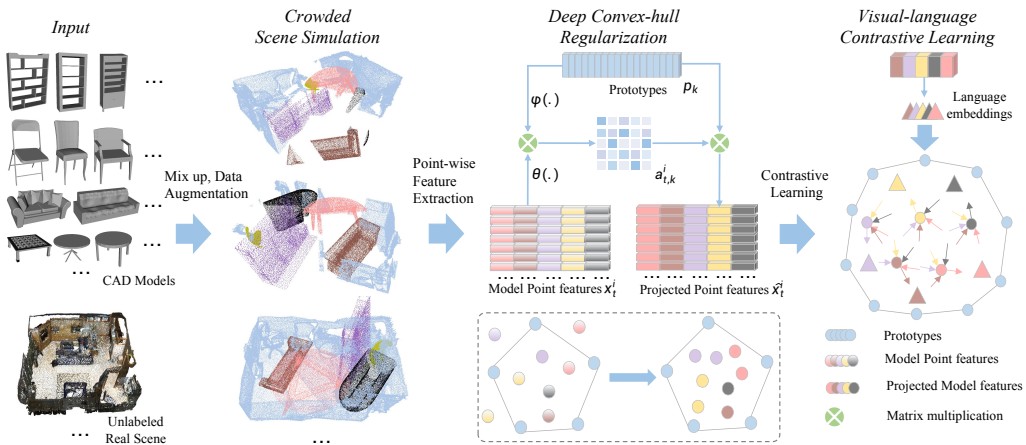

Figure 2: The framework of Model2Scene in the training stage. Firstly, we simulate the crowded scene by mixing up the CAD models with data augmentation, including random rotation, scaling, cropping, and mixing up with the scene (these coloured points are from CAD models). Secondly, we extract the point-wise feature from the simulated crowded scene and project the point features into a convex hull space via Deep Convex-hull Regularization, where the space is surrounded by a group of learned prototypes. In the end, we perform visual-language contrastive learning to align the projected point features and the language embeddings.

## 2 MODEL2SCENE

**Problem Definition** Give 3D CAD models $\{M_i\}_{i=1}^{N^m}$ with labels $\{G_i\}_{i=1}^{N^m}$, we aim to learn 3D scene representation from the 3D CAD models and evaluate the scene perception performance on real scene scans $\{S_j\}_{j=1}^{N^s}$. $N^m$ and $N^s$ are the number of models and scenes, respectively. To efficiently learn the 3D scene representation from individual CAD models, the main challenge is solving the model-to-scene (from a single model to the scene) and synthetic-to-real (from CAD model to real scene's object) gap between the CAD model and the real scene's objects. In the training stage, the input is labelled CAD models, while the input is only a scene scan in the testing stage. We conduct several downstream tasks to evaluate the learned 3D scene representation, including 3D object saliency detection, label-efficient 3D perception and zero-shot 3D semantic segmentation.

**Approach Overview** As illustrated in Fig. 2, Model2Scene consists of three modules. Firstly, we mix up the CAD models to simulate a crowded scene that reduces the model-to-scene gap. Secondly, a novel Deep Convex-hull Regularization is proposed, in which we map the point features into a unified convex hull space surrounded by a group of learned prototypes. In the end, we perform contrastive learning on the mapping features with the language semantic embeddings. The cooperation of these steps leads to the success of 3D scene representation learning from the CAD models. In what follows, we will present these components in detail.

### 2.1 CROWDED SCENE SIMULATION

Simulating a crowded scene using CAD models is an intuitive and straightforward solution to ease the model-to-scene gap. Specifically, our first step is to unify the data format, *i.e.*, CAD models are presented in Mesh format that consists of the vertexes and faces. We transfer the CAD model mesh to a uniform point cloud by Poisson Disk Sampling (Yuksel, 2015), and ensure the density closer to the scene scan. In the next step, we randomly place the CAD models on the scene floor (regardless of the layout), with or without filtering the overlapped points. Besides, inspired by current data augmentation methods, a series of data processing approaches are introduced to cope with the CAD models to cover the diversity of the objects in a real scene scan, including scaling, rotation, and Cropping. Specifically, we randomly scale CAD models to roughly match the real scene's object size (not model fitting). Besides, random rotation transformation is also applied to capture the pose diversity of an object. Finally, considering the object in a scene scan is always partially observed, a random cropping strategy is designed to simulate this scenario, *i.e.*, we first randomly sample 2∼5 points from the model as anchor points and then cluster all points based on their Euclidean distance to the anchor points. During training, one of a cluster will be randomly filtered.

Figure 3: The sub-picture A is the framework in the inference stage. The sub-picture B is the visualization of two learned prototypes (from left to right are plane and hole structure, respectively).

## 2.2 DEEP CONVEX-HULL REGULARIZATION

Since the network is only trained on the CAD models, its feature space typically differs from the object in a real scene scan, leading to low inferring accuracy. Inspired by the convex hull theory (Rockafellar, 2015), we propose a novel Deep Convex-hull Regularization (DCR) to project point features into a convex hull space for eliminating the domain gaps. In what follows, we revisit the convex hull theory and present our DCR in detail.

**Revisiting Convex Hull Theory**    Convex hull is a fundamental concept in computational geometry. It is defined as the set of all convex combinations, where the convex combination is a linear combination of the support points, and all coefficients are non-negative and sum to 1. In conclusion, if a combination is a convex combination, it must remain in the convex hull. Therefore, regarding the point features as a convex combination, we could restrict all point features from different domains into a unified feature space, thus alleviating the domain gap.

**Formulation**    We set a group of learnable prototypes $\{p_k\}_{k=1}^{K}$ as the support points of a convex hull, where $p_k \in \mathbb{R}^D$ and $K > D$. $K$ denotes the number of prototypes, and $D$ is the dimension of a prototype. Note that prototypes are automatically updated via back-propagation. What is interesting is that the prototypes learn the base structural elements of 3D models (Fig. 3 (B)). In this context, a point feature can be formulated as a linear combination of the related structural elements.

Given the point features $\{x_t^i\}_{t=1}^{T}$ with $x_t^i \in \mathbb{R}^D$ extracted by the encoder $E$ from the $i$-th CAD model with $T$ points, the corresponding mapping feature $\{\hat{x}_t^i\}_{t=1}^{T}$ is obtained by the following function.

$$\hat{x}_t^i = \sum_{k=1}^{K} a_{t,k}^i * p_k, \sum_{k=1}^{K} a_{t,k}^i = 1, \tag{1}$$

where $a_{t,k}^i$ serves as the coefficient to the corresponding prototype, defined by:

$$a_{t,k}^i = \exp(\lambda * d(\theta(x_t^i), \varphi(p_k)))/\Gamma,$$
$$\Gamma = \sum_{k=1}^{K} \exp(\lambda * d(\theta(x_t^i), \varphi(p_k))), \tag{2}$$

where $d(\cdot)$ measures the similarity between the point feature and the $k$-th prototype, which is dot product operation in this work. $\theta(\cdot)$ and $\varphi(\cdot)$ denote the key and the query function (Vaswani et al., 2017), respectively. $\lambda$ is the inversed temperature term (Chorowski et al., 2015).

Essentially, the feature embedding $\hat{x}_t$ is a convex combination of the prototypes due to the coefficients $a_{t,k}^i > 0$ and sum to 1. Therefore, $x_t$ is mapped into a convex $\mathbb{W} \subset \mathbb{R}^D$, where $\mathbb{W}$ is a closure and compact metric space surrounded by the learned prototypes.

In the inferring phase (Fig. 3 (A)), the point feature $x_t \in \mathbb{R}^D$ from a real scan first accesses the most relevant prototypes to obtain the coefficients. Then, it is transferred to a mapping feature $\hat{x}_t \in \mathbb{W}$ which is the convex combination of the prototypes. In this way, the feature space of the CAD models and the real scene's objects are projected to the unified subspace $\mathbb{W}$ surrounded by a group of learned prototypes $\{p_k\}_{k=1}^{K}$, offering the network a better generalization ability.

## 2.3 VISUAL-LANGUAGE CONTRASTIVE LEARNING

As all point features are projected to the convex hull subspace $\mathbb{W}$, we cluster these points to ensure they are inner-class compact and inter-class distinguishable. We introduce the language embeddings

Table 1: Evaluation on the ScanNet. MinkUNnet is the baseline method. We apply point-wise feature adaptation in ADDA and instance adaptation in ADDA†. 'Supervised' indicates training with the point-wise annotations.

| Method | AmAP | chair | table | bed | sink | bathtub | door | curtain | desk | bookshelf | sofa | toilet |
|---|---|---|---|---|---|---|---|---|---|---|---|---|
| Baseline | 10.93 | 8.09 | 10.60 | 15.97 | 2.53 | 12.40 | 9.77 | 13.92 | 4.66 | 26.76 | 11.49 | 4.07 |
| ADDA (Tzeng et al., 2017) | 28.93 | 29.93 | 40.88 | 36.82 | 8.03 | 31.60 | 13.59 | 25.10 | 20.01 | 35.49 | 33.90 | 42.92 |
| ADDA† (Tzeng et al., 2017) | 38.14 | 67.03 | **48.60** | 29.77 | 20.36 | 29.70 | 12.04 | 23.06 | 26.57 | 42.69 | 51.41 | 68.31 |
| PointDAN (Qin et al., 2019) | 32.92 | 58.31 | 40.39 | 20.96 | 12.58 | 31.65 | 11.79 | 17.65 | 26.04 | 48.31 | 41.25 | 53.18 |
| 3DIoUMatch (Wang et al., 2021) | 42.25 | 63.34 | 41.24 | 43.58 | 26.53 | **46.61** | 15.52 | 24.45 | 26.75 | 45.49 | 54.49 | **76.81** |
| Ours | 46.08 | **67.19** | 46.73 | **45.65** | **31.28** | 43.36 | **17.47** | **34.43** | **29.15** | **59.37** | **60.48** | 71.77 |
| Supervised | 78.98 | 91.44 | 72.89 | 76.35 | 83.94 | 84.84 | 58.47 | 74.22 | 72.96 | 75.76 | 84.81 | 93.10 |
| Supervised+ours | 79.46 | 92.16 | 74.37 | 76.50 | 85.18 | 85.09 | 56.57 | 70.68 | 73.54 | 77.03 | 87.44 | 95.52 |

Table 2: Evaluation on the S3DIS area 5 test. MinkUNnet is the baseline method. We apply point-wise feature adaptation in ADDA and instance adaptation in ADDA†. 'Supervised' indicates training with annotated scenes.

| Method | AmAP | chair | bookshelf | sofa | table |
|---|---|---|---|---|---|
| Baseline | 15.65 | 3.75 | 37.72 | 13.34 | 7.80 |
| ADDA (Tzeng et al., 2017) | 26.57 | 26.67 | 54.70 | 15.91 | 9.00 |
| ADDA† (Tzeng et al., 2017) | 30.04 | 57.80 | 54.70 | 22.91 | 8.70 |
| PointDAN (Qin et al., 2019) | 39.80 | 56.60 | 52.34 | 32.06 | 18.20 |
| 3DIoUMatch (Wang et al., 2021) | 49.60 | 69.07 | 56.87 | **54.25** | 18.21 |
| Ours | **55.49** | **70.86** | **62.68** | 47.87 | **40.57** |
| Supervised | 90.44 | 97.05 | 87.84 | 86.83 | 90.06 |
| Supervised+ours | 92.57 | 97.83 | 89.86 | 92.05 | 90.52 |

$\{h_c\}_{c=1}^C$ with $h_c \in \mathbb{R}^D$ to indicate the clustering centres in metric space, where the language embeddings are the output embeddings of the word2vec (Mikolov et al., 2013) or glove (Pennington et al., 2014) model with the input of categories' names. Given the point features $\{\hat{x}_t^i\}_{i=1,t=1}^{N^m,T}$ from $N^m$ CAD models$\{M_i\}_{i=1}^{N^m}$, we pull in whose points to the corresponding language embeddings while pushing away from the rest of language embeddings according to their semantic labels $\{G_i\}_{i=1}^{N^m}$. Therefore, the points are compact for inner class and distinguishable for inter classes in the metric space $\mathbb{W}$. For simplicity, we adopt Cross-Entropy loss in this paper.

$$\mathcal{L} = -\log \sum_{i=1}^{N^m} \sum_{t=1}^{T} \frac{\exp(d(\hat{x}_t^i, h_{G_i}))}{\sum_{c=1}^{C} \exp(d(\hat{x}_t^i, h_c))}, \tag{3}$$

When inferring an unseen scene scan $S_j$ with the point features $\{\hat{x}_t^j\}_{i=1}^T$, the possibility distribution of the $t$-th point that belongs to each class is determined by the similarities between the point feature $\hat{x}_t^j$ and the class-specific anchors $\{h_c\}_{c=1}^C$ (Fig 3 (A)).

$$l_t^c = \exp(d(\hat{x}_t^j, h_c))/\gamma, \gamma = \sum_{c=1}^{C} \exp(d(\hat{x}_t^j, h_c)), \tag{4}$$

where $l_t^c$ is the possibility that the $t$-th point belongs to the $c$-th class, $\gamma$ is a normalization term.

## 3 EXPERIMENTS

### 3.1 DATASET

We conduct the experiments on ModelNet40 (Wu et al., 2015), ScanNet (Dai et al., 2017) and S3DIS (Armeni et al., 2017) datasets, where ModelNet provides the CAD model for training, and the scene scans in ScanNet and S3DIS are for evaluation.

**ModelNet** is a comprehensive clean collection of 3D CAD models for objects, composed of 9843 training models and 2468 testing models in 40 classes. We transfer the model mesh to 8196 uniform points by Poisson disk sampling (Yuksel, 2015). We take the 9843 models from the training set as the CAD models in our method. **ScanNet** contains 1603 scans, where 1201 scans for training, 312 scans for validation and 100 scans for testing. The 100 testing scans are used for the benchmark, and their

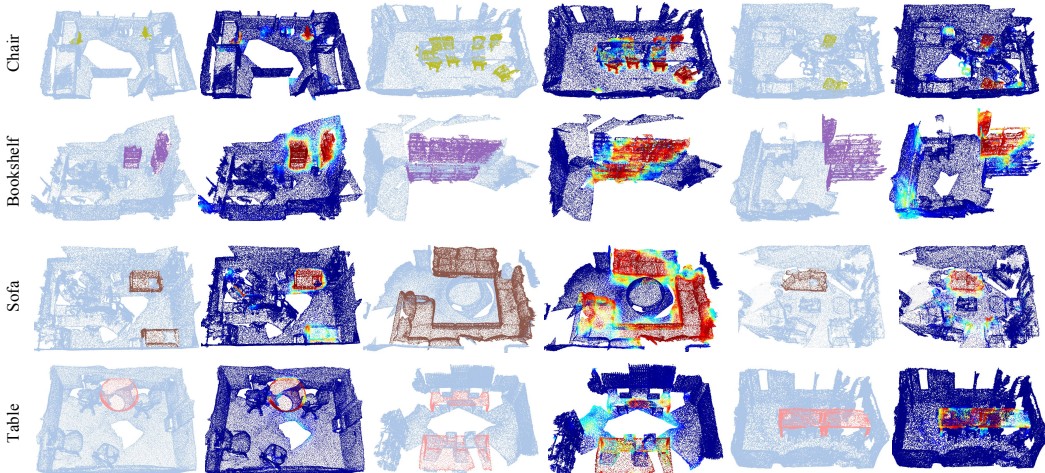

Figure 4: Visualization of 3D object saliency detection on the ScanNet dataset. We show the pairs of ground truth (left) and the inferring results (right). From up to down are chair, bookshelf, sofa and table, respectively.

labels are unaccessible. There are 11 identical classes to the ModelNet40 dataset, including the chair, table, desk, bed, bookshelf, sofa, sink, bathtub, toilet, door, and curtain. We take the 1201 scans to mix up with the synthetic models for training (labels are not used), and the rest of the 312 scans are used to evaluate the performance. **S3DIS** consists of 271 point cloud scenes across six areas for indoor scene semantic segmentation. There are 13 categories in the point-wise annotations, where four identical classes to the ModelNet40 dataset, including the chair, table, bookcase (bookshelf) and sofa. We utilize Area 5 as the validation set and use the other five areas as the training set (labels are not used), the same with the previous works (Li et al., 2018; Qi et al., 2017; Jiang et al., 2021)

## 3.2 EVALUATION METRIC

As for 3D object saliency detection, the goal is to detect the objects (point clouds) that belong to the identical class with CAD models. Therefore, we calculate the class-specific point-wise possibility on the scene scan and adopt the mean Average Precision (mAP) to measure the performance for each class. AmAP is the average mAP of all classes. We believe AmAP is more suitable than mIoU because the foreground category of objects occupies a small proportion in a whole scene.

## 3.3 IMPLEMENTATION DETAILS

We adopt MinkowskiNet14 (Choy et al., 2019) as the backbone to extract the point-wise feature. Thus, the feature dimension $D$ is set to be 96. The key $\theta(\cdot)$ and the query $\varphi(\cdot)$ function are linear transformation and output 16-dimensional vectors. The voxel size of all experiments is set to be 5 cm for efficient training. Our method is built on the Pytorch platform, optimized by Adam with the default configuration. The batch size for the ModelNet, ScanNet and S3DIS are 4 * $(Q + 1)$, 4 and 4, respectively, indicating that one scan is mixed up with $(Q +1)$ synthetic models, where $Q$ is the number of identical classes between two datasets and one for the negative sample. Since there is no colour in the synthetic models, we set the feature in the ScanNet and S3DIS dataset to be a fixed tensor (1), identical to that in the ModelNet. Training 200 epochs costs 15 hours on two RTX 2080 TI GPUs. During training, we randomly rotate the models and scans along the z-axis, randomly scale the model and scene with scaling factor 0.9-1.1 and randomly displace the model's location within the scene. If there are overlapped points, we randomly filter or maintain them. We take all identical classes as foreground classes to evaluate the performance and utilize the remaining classes as negative samples for contrastive learning. More details are in supplementary materials.

## 3.4 RESULTS AND DISCUSSIONS

In this section, we report the performance of three downstream tasks: 1. label-free 3D object salient detection; 2. label-efficient 3D perception; and 3. zero-shot 3D semantic segmentation.

Table 3: Fine-tuning on the Scannet and S3DIS dataset for semantic segmentation task. The number in () donates the improved accuracy compared with purely supervised training.

| | Scannet | | | S3DIS | |
|---|---|---|---|---|---|
| | 5% data | 10% data | 100% data | MinkNet14 | MinkNet34 |
| Trained from scratch | 50.24 | 54.86 | 63.05 | 56.44 | 58.63 |
| PointContrast (Xie et al., 2020) | 55.31(5.07) | 58.68(3.82) | 65.03(1.98) | 58.65(2.21) | 60.71(2.08) |
| Ours | **56.46(6.22)** | **59.17(4.31)** | **65.14(2.09)** | **58.94(2.50)** | **60.79(2.16)** |

Table 4: Fine-tuning on the Scannet for 3D object detection results. The number in () donates the improved accuracy compared with fully supervised training.

| Model | mAP@0.5 | mAP@0.25 |
|---|---|---|
| Trained from scratch | 31.82 | 53.39 |
| PointContrast (Xie et al., 2020) | 34.30(2.48) | 55.56(2.17) |
| Ours | **34.51(2.69)** | **55.60(2.21)** |

### 3.4.1 LABEL-FREE 3D OBJECT SALIENT DETECTION

**Baselines** No deep learning-based methods have investigated this problem to the best of our knowledge. Therefore, we build a baseline method (baseline in Table 2 and Table 1) without Crowded Scene Simulation (CSS) and Deep Convex-hull Regularization (DCR). Specifically, we first resize the CAD models to the same scale as the scene's objects, then extract the point feature for individual models and classify the points with the model labels. Besides, to verify the superiority of Deep Convex-hull Regularization, we compare it with a semi-supervised method (3DIoUMatch (Wang et al., 2021)) and two unsupervised domain adaptation methods (ADDA (Tzeng et al., 2017) and PointDAN (Qin et al., 2019)). Specifically, the stimulated crowed scene is regarded as labelled data in 3DIoUMatch, and as the source domain in ADDA and PointDAN. Note that to adapt 3DIoUMatch for the semantic segmentation task, we calculate the mask IoU instead of the bounding box IoU.

**Results** As shown in Table 1 and 2, our method achieves 46.08% AmAP and 55.49% on the **ScanNet** and **S3DIS** dataset, which significantly outperforms other methods. Compared with the baseline, indicating the effectiveness of Model2Scene. Furthermore, the Deep Convex-hull Regularization is verified to be feasible as it achieves a better performance than 3DIoUMatch, ADDA and PointDAN. We also show the performance by training on the annotated scene scans (Supervised). When training on both CAD models and annotated scene scans (supervised+Ours), the performance is higher than that only training on ground truth. The qualitative evaluation is shown in Fig. 4. More results are in supplementary materials.

### 3.4.2 LABEL-EFFICIENT 3D PERCEPTION

The learned 3D scene representation is beneficial for the downstream tasks. We fine-tune the network with different proportions of labelled scans for semantic segmentation on the ScanNet dataset. For the S3DIS dataset, we evaluated two backbone networks (MinkowskiNet14 and MinkowskiNet34) for semantic segmentation. As shown in Table 3, our method outperforms Pointcontrast (Xie et al., 2020). Note that we compare the original version of Pointcontrast without leveraging additional models. Compared with purely supervised counterparts, a significant improvement could be observed in the two datasets for both seen and unseen categories (not shown on synthetic models). Besides, our method is also beneficial for the object detection task (Table 4). More experiment results are in supplementary materials.

### 3.4.3 ZERO-SHOT 3D SEMANTIC SEGMENTATION

As the point feature is aligned with language embedding, the network is capable of zero-shot ability. We evaluate the performance in the ScanNet dataset. Seen classes are those overlapped classes in the ScanNet and ModelNet datasets, while unseen classes are the rest of the classes in the ScanNet dataset. The results are shown in Table 5.

Table 5: Zero-shot semantic segmentation on scannet. mIoU is the metric.

| Model | All classes | Seen classes | Unseen classes |
|---|---|---|---|
| Ours | **13.41** | **21.01** | **4.11** |

Table 6: Ablation experiments. MinkUNnet is the baseline method. MMS and FA donate Models and Scene Mix-up and convex-hull regularized feature alignment, respectively.

| Method | AmAP | chair | table | bed | sink | bathtub | door | curtain | desk | bookshelf | sofa | toilet |
|---|---|---|---|---|---|---|---|---|---|---|---|---|
| Base | 10.93 | 8.09 | 10.60 | 15.97 | 2.53 | 12.40 | 9.77 | 13.92 | 4.66 | 26.76 | 11.49 | 4.07 |
| Base+CSS$_{noDA}$ | 23.44 | 36.90 | 35.57 | 26.30 | 11.98 | 17.29 | 11.01 | 21.07 | 11.58 | 25.03 | 16.40 | 44.66 |
| Base+CSS$_{negaSc}$ | 23.04 | 41.74 | 25.05 | 21.92 | 7.86 | 23.01 | 13.88 | 21.93 | 14.79 | 23.80 | 13.08 | 46.39 |
| Base+CSS$_{negaMo}$ | 41.34 | **73.12** | 40.05 | 42.73 | 29.17 | 19.97 | 17.52 | 23.33 | 33.40 | 50.34 | 59.36 | 65.71 |
| Base+CSS$_{negaMo\_noCo}$ | 37.83 | 72.62 | 39.61 | 37.49 | 14.95 | 16.93 | 18.33 | 21.29 | 32.84 | 46.09 | 59.44 | 56.53 |
| Base+CSS | 42.19 | 67.53 | 47.64 | 41.82 | 27.01 | 32.47 | 13.68 | 27.82 | 26.78 | 50.54 | 58.75 | 70.07 |
| Base+CSS+DCR$_{K64}$ | 43.16 | 67.37 | **48.27** | 41.51 | 20.38 | 33.90 | 16.30 | 31.55 | 27.17 | 52.32 | **61.19** | 74.84 |
| Base+CSS+DCR$_{K128}$ | **46.08** | 67.19 | 46.73 | 45.65 | **31.28** | 43.36 | 17.47 | 34.43 | 29.15 | **59.37** | 60.48 | 71.77 |
| Base+CSS+DCR$_{K256}$ | 43.81 | 68.15 | 45.12 | **51.08** | 20.12 | 38.14 | 17.10 | 25.64 | **33.99** | 48.23 | 59.32 | 75.05 |
| Base+CSS+DCR$_{T4}$ | 44.51 | 69.05 | 47.28 | 49.72 | 18.26 | **45.30** | **18.46** | 28.83 | 26.16 | 49.94 | 57.23 | **79.40** |
| Base+CSS+DCR$_{T0.1}$ | 43.79 | 70.47 | 42.44 | 44.26 | 20.95 | 39.57 | 16.00 | 29.92 | 27.65 | 52.04 | 59.14 | 79.20 |
| Base+CSS+DCR$_{cos}$ | 43.45 | 69.16 | 45.84 | 42.53 | 21.47 | 37.38 | 15.36 | 29.20 | 29.06 | 51.72 | 59.16 | 77.06 |

## 3.5 ABLATION STUDY

We evaluate the performance of 3D object saliency detection on ScanNet to verify the effectiveness of different modules, including Crowded Scene Simulation (CSS) and Deep Convex-hull Regularization (DCR). In the following, we present the configuration details and give more insights into what factors affect the performance.

**Effect of Crowded Scene Simulation** Base+CSS donates the baseline with the Crowded Scene Simulation (CSS). Observing from (Base, Base+CSS), AmAP is improved by 31.26%. We dig into CSS by exploring the following configurations. Firstly, we investigate how data augmentation (DA) influences performance, including random scaling and rotation. These operations cover the diversities of the scene's objects. As shown in Table 6, the performance greatly reduced if without applying DA (Base+CSS$_{noDA}$ (23.44 AmAP) VS Base+CSS (42.19 AmAP)). Besides, we find the random cropping is beneficial for performance improvement due to the real scene's objects are often partially scanned (Base+CSS$_{negaMo\_noCo}$ (37.83 AmAP) is without random cropping). Secondly, to explore how to conduct negative samples, we try to take the scene's points as the negative samples for contrastive learning (Base+CSS$_{negaSc}$). We find that the performance (23.04 AmAP) is significantly worse than Base+CSS (42.19 AmAP), which uses the CAD models as negative samples. It is probably that the network learns the artefacts to distinguish the CAD models from the scene. The artefacts are mainly caused by the mixing up operation, such as the overlapped/disjointed points. As a result, the network could not generalize the knowledge to a clear scene without such artefacts. Lastly, to understand the role of mixing the scene, we only mix up the CAD models together and exclude the scene points (Base+CSS$_{negaMo}$). Surprisingly, the performance is comparable with the counterpart Base+CSS (41.34 AmAP VS 42.19 AmAP).

**Effect of Deep Convex-hull Regularization** Since feature domain gaps exist between the CAD models and the objects in a real scan, we use the prototypes to align their features into an unified feature space (Base+CSS+DCR). The experiment shows that the improvement is about 4% for AmAP (Base+CSS VS Base+CSS+DCR$_{K}$128). The inversed temperature $\lambda$ and the number of prototypes $K$ are two hyper-parameters for the feature alignment module. $\lambda$ indicates the smoothness of the coefficient distribution and can be regarded as a regularization term to prevent network degradations. We present the results when $\lambda$ is 0.1, 0.5 and 4, respectively. (Base+CSS+DCR$_{T0.1}$, Base+CSS+DCR$_{K128}$ and Base+CSS+DCR$_{T4}$). We choose $\lambda$ to be 0.5 empirically. The number of prototypes $K$ is another hyper-parameter. We respectively evaluate it with the configurations Base+CSS+DCR$_{K64}$, Base+CSS+DCR$_{K128}$, and Base+CSS+DCR$_{K256}$. The network achieves the best performance when $K$ is set to be 128. Besides, we show the result when the key $\theta(\cdot)$ and the query function $\varphi(\cdot)$ are identity mapping function in the setting Base+CSS+DCR$_{cos}$. The performance is slightly worse than the full method Base+CSS+DCR$_{K128}$.

## 4 CONCLUSION

We propose Model2Scene to investigate the problem of learning 3D scene representation from CAD models and languages. Notably, we propose a novel Deep Convex-hall Regularization to handle the model-to-scene and synthetic-to-real domain gaps. Extensive experiments show that the learned 3D scene representation can favour the downstream task's performance, including label-free 3D salient object detection, label-efficient 3D perception and zero-shot 3D semantic segmentation.

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
