# OpenReview forum: "Model2Scene: Learning 3D Scene Representation via Contrastive Language-CAD Models Pre-training"
_ICLR.cc/2024/Conference — ICLR 2024 Conference Withdrawn Submission_

### Official Review · Reviewer_U8AX · 2023-10-29

**Soundness:** 3 good
**Presentation:** 2 fair
**Contribution:** 3 good
**Rating:** 6
**Confidence:** 3

**Summary:**

Authors proposed Model2Scene framework for learning 3D scene representation from CAD models and languages. After data-augmentation of CAD models, a deep convex-hull regularization maps point features as a convex combination of anchor features, and contrastive loss based on language embedding is further used to enhance the intra/inter class variance. Authors demonstrated good results on ScanNet and SDIS.

**Strengths:**

Results are very strong. Improvements on multiple tasks across ScanNet and S3DIS clearly demonstrate the effectiveness of deep convex-hull regularization and language-cad pretraining. Authors also show that features learned by Model2Scene on only CAD data can transfer well to the real world without supervised learning, which is a big improvement over existing works.

**Weaknesses:**

Writing lacks details for the training and inference stages. Details such as how the prototypes are learned via back-propagation should be explained more. The pipeline in figure 3 is also confusing, especially the sub-picture A and B on the right.

I also have some concerns about the modules used in model2sence. For data augmentation, I am not sure why authors go with random position and scaling of CAD models. Scene differs from a single object in that it is made up of several objects following a certain room layout. It seems weird to fully ditch the layout information, especially for 3D scene representation learning.

The deep convex hull regularization is similar to existing anchor-based point feature learning methods, which has also been applied to scene-level and demonstrated good results. And this leads to another issue with baseline comparison methods. For the results in table 1 and 2, all baselines are trained at the most basic point-level. More advanced baselines such as those anchor-based ones should be included for fairness.

**Questions:**

As explained in the weakness, authors should provide more details for training and inference. Also although the result improvements are good, the understanding of why things work, especially for deep hull regularization and data augmentation are not very well explained. Baseline comparisons to prove the effectiveness of deep hull regularization also seems weak.

---

### Official Review · Reviewer_xgfx · 2023-10-30

**Soundness:** 2 fair
**Presentation:** 1 poor
**Contribution:** 1 poor
**Rating:** 3
**Confidence:** 4

**Summary:**

The paper presents a framework for learning 3D indoor scene representations from CAD models and language commands. The goal is to learn better scene representation for scene understanding tasks, mainly perception tasks such as 3D object detection and scene segmentation. There exist two main challenges in solving this task using a dataset of CAD models – (1) the existence of CAD_models-to-3D_scenes domain gap, and (2) synthetic to real domain gap. It should be noted that the 3D indoor scenes are represented as point clouds in this work.

These challenges are addressed by three main parts: data pre-processing, latent space regularization using Deep Convex Hull regularization, and designing an objective function that allows for using language embeddings to better learn point feature embeddings in the 3D scene. Note that the points can be sampled on the surface of a CAD model (see below).
The data pre-processing step is done to reduce the model-to-scene gap. Here, CAD models are placed on the floor of an existing scanned scene (represented as a point cloud) to simulate a “crowded” scene (as is referred to in the paper; see P2, second line in the first paragraph). The CAD models and the 3D scene scans are rotated randomly, scaled randomly by a scaling factor in the range [0.9, 1.1], and a model is randomly placed in a scene.
The latent regularization step is done by projecting point features from the input CAD models and the input scene into a common space constrained to be convex. This step is called a Deep Convex Hull Regularization (DCR).

To summarize, Input:  Labeled CAD models and a 3D scene scan (i.e., a point cloud)
Task: Learn scene representations

Dataset used:
ModelNet , ScanNet and S3DIS datasets

Underlying Neural Network:
The paper uses two types of neural frameworks.
One is MinkowskiNet14 whose backbone is made of 3D Conv Layers. The other framework that is employed is a MLP-based contrastive learning setting (this is where visual-language connection is introduced).

Loss function:
Cross Entropy Loss in learning the visual-language embeddings (see Eq 3)

Quantitative Metric:
3D object saliency detection -- mAP (mean average precision),
Zero-shot 3D semantic segmentation – mIoU

**Strengths:**

1)	The paper provides experiments on real scene datasets to better evaluate the proposed framework.
2)	Ablation studies seem to be thorough.

**Weaknesses:**

1)	The paper is poorly written, especially the Abstract, Intro, and parts of the Method Sections. For example, in Sec 2.2, the first line says “Since the network is only trained….” – there has been no mention of the “the” network before this in the Method section. The clarity of thought in writing this is lacking. The readers can understand that there is a neural network that is employed but the first sentence does not document the research well. Please provide a context first and then follow it up with the first sentence of Sec 2.2.
The same goes for the first reference of DCR on P2. It is difficult to understand what point features are referred to in this sentence “to project point features into a unified convex hull space that further ….”. Prior to this sentence, there is no info/mention on what kind of representation is used. Okay, the reader has to understand that it is point cloud representation. But which specific points are being sampled/referred to? You get my point. The paper lacks serious clarity of writing. This is also seen in the Section numbering, where the word “Introduction” as a section heading is missing.

2)	I am not sure what a “crowded scene” means, but I am guessing the paper meant to say that it is to simulate a real-world scene. This, as is claimed in the paper, is to reduce the CAD_model-to-scene gap. I agree to some extent on this, but it can also reduce the synthetic-to-real gap. Why is this not the case?
3)	It would help the readers if a single sentence is written to introduce “prototypes” before starting with the formulations on P5. I am still confused as to what is a “prototype”. Is it a template/basis function?
4)	It is unclear as to why there is a need to introduce visual-language embeddings. I am having a hard time understanding the motivation for going from the DCR step to the language embedding step. Moreover, it is not described as to how these language prompts for the CAD models/ scenes are obtained. The overall framework of the paper is not convincing.
5)	Fig 4 – what is the notion of saliency? Is it just object detection? If so, how does the algorithm understand saliency in a scene composed of multiple objects? Plus, how is the saliency visualized?
6)	Not sure if PointContrast is the only work that needs to be compared to. There are follow-ups of PointContrast that are more recent, which could be discussed. This is a weak weakness, so, I can overlook this if the paper fared in other aspects.
7)	No discussion of the limitations

**Questions:**

1)	Table 3 – This table represents quant results for semantic segmentation on the S3DIS dataset. The caption says “improved accuracy”, but as I understand, the evaluation metric for semantic segmentation is mIoU. There is a discrepancy here. Using the word “accuracy” is misleading.
2)	The same goes for Table 4, but it is a little clearer in this Table that “accuracy” is being referred to as mAP scores. Again, using a better term, like “improved performance”, is apt.
3)	As explained in Sec 2.1, a CAD model is placed on the scene floor. How is this done? Can you briefly discuss this?
4)	Also, it is mentioned that the CAD models are scaled to match real scenes’ object size. I am not sure how this is done. The scene scan contains objects of different sizes, say, TV stand, Lamp, Coffee Table, etc. and as I understand, these scene semantics are not known. Only the CAD model labels are known during training. So, how is the scaling done? One needs to have a reference object (i.e., its point cloud) from the scene to obtain the scaling factor by which the CAD model needs to be scaled when it is being placed in the scene.
5)	Random rotation transformation on CAD models – This cannot be true. If you do this in the SO3 space, rotated objects will be implausibly placed in the scene.

---

### Official Review · Reviewer_yhuq · 2023-10-31

**Soundness:** 2 fair
**Presentation:** 2 fair
**Contribution:** 2 fair
**Rating:** 5
**Confidence:** 3

**Summary:**

In this paper, the authors propose to learn 3D scene representation from CAD models and languages. The author raises two challenges in adapting CAD models to real 3D scenes: model-to-scene and synthetic-to-real. To solve the model-to-scene problem, the authors propose to place objects into crowded scenes as scene simulation. To solve the synthetic-to-real problem, the authors propose a deep convex-hull regularization to update point features. Visual-language contrastive learning is utilized as loss function to optimize the pipeline.

**Strengths:**

- The paper is well organized. The model-to-scene and synthetic-to-real problem is adequately extracted and the proposed model matches perfect with the problems.

**Weaknesses:**

- Why the proposed deep convex-hull regularization can solve the synthetic-to-real problem? The authors are encouraged to discuss the underlining reasons since there are no supervision from the real-world to guarantee that the features are adapted to the real space.
- The authors are encouraged to compare the proposed method with more advanced and recent literature to make the results more convincing
- The figures in this paper are not well presented. For example, the sub-picture A/B in Figure 3 is not clearly noted.
- The ablation table is not well organized and is confusing.

**Questions:**

See above

---

### Official Review · Reviewer_gV7n · 2023-10-31

**Soundness:** 2 fair
**Presentation:** 2 fair
**Contribution:** 2 fair
**Rating:** 3
**Confidence:** 3

**Summary:**

This paper mainly designed a data augmentation paradigm that simulated a crowded scene by mixing data-augmented CAD models. Coupling with a feature regularization operation, termed Deep Convex-hull Regularization (DCR) tries to solve an annotation-poor problem of data in 3D perception. This is an interesting problem worth investigating.

**Strengths:**

Good data augmentation paradigm for point cloud perception, only with a synthesized CAD dataset, but achieving good performance on real-world datasets like ScanNet et al.

**Weaknesses:**

Writing:
After reading the whole paper, the main contribution of this paper is the paradigm of data augmentation that partially alleviates the problem of annotation-poor in 3D perception community. Its ok, and worth investigating. However, the writing does not precisely convey the real contribution.  For example, the second paragraph of the Introduction, makes me feel like this work solves all issues in this community.

Experiment:
For the choice of baseline, the author claimed "No deep learning-based methods have investigated this problem to the best of our knowl- edge."  (see the first sentence of section 3.4.1) It doesn't make sense to me. Dozens of Domain Adaptation methods could be your baseline methods.

**Questions:**

1. From my understanding, most 3D augmentation methods (as you mentioned in the related work section), and domain adaptation methods should be as baselines to be compared. Could you give me reasons why you didn't do that?

2. Try your augmentation method with other supervised methods to see if it could be a general data augmentation that benefits most of them.

3. Please correct me if any misunderstandings in my comments, I am happy to discuss on them.